# The Impact of Age, Gender and Technical Experience on Three Motor Coordination Skills in Children Practicing Taekwondo

**DOI:** 10.3390/ijerph18115998

**Published:** 2021-06-03

**Authors:** Stefanos Boutios, Giovanni Fiorilli, Andrea Buonsenso, Panagiotis Daniilidis, Marco Centorbi, Mariano Intrieri, Alessandra di Cagno

**Affiliations:** 1Department of Physical Education and Sport Science, National and Kapodistrian University of Athens, 157 72 Athens, Greece; 2Department of Medicine and Health Sciences, University of Molise, v. De Sanctis 1, 86100 Campobasso, Italy; fiorilli@unimol.it (G.F.); andreabuonsenso@gmail.com (A.B.); marco.centorbi@hotmail.it (M.C.); intrieri@unimol.it (M.I.); 3Department of Physical Education and Sport Science Serres, Aristotle University of Thessaloniki, 541 24 Thessaloniki, Greece; pandanproteas@yahoo.gr; 4Department of Motor, Human and Health Sciences, University of Rome “Foro Italico”, Lauro de Bosis Square 15, 00197 Rome, Italy; alessandra.dicagno@uniroma4.it

**Keywords:** martial arts, maturation, gender differences, reactive time

## Abstract

The study aim was to investigate the age, gender and technical level on motor coordination abilities of Taekwondo children. One hundred and fifteen children (83 male, 32 female), aged 7.76 ± 1.71 years, divided in three different groups, under 8 (5–7 years), under 10 (8–9 years) and under 12 (10–11 years), underwent three coordination skills tests: the ruler drop test (RDT), assessing visual reaction time, the hexagonal test (HT), assessing agility, and the target kick test (TKT), assessing kicking ability. MANOVA showed significant gender differences for TKT, in which females showed higher scores than males (*p* = 0.033). Significant differences were found in HT and TKT, where the under 12 group showed higher scores than younger athletes (*p* < 0.001). No differences amongst different age groups were found in RDT, showing that this could be a good predictor of Taekwondo performance, assessed at an early stage. High-level athletes showed better scores in all the tests than the low levels, as it was expected. Coordinative performance improves with age and is positively influenced by practicing a sports activity. The predisposition to a particular sport with a well-planned training may lead to a motor proficiency comparable to that reached by older athletes and better than same-age athletes.

## 1. Introduction

Childhood is the critical time for learning and strengthening the fundamental motor skills, considering that motor skill proficiency is positively associated with health, fitness and academic outcomes [1]. The acquisition of coordination and motor competence is not only achieved through maturation, but also through continuous interaction with a stimulating and supportive social and physical environment.

The practice of taekwondo provides improvement in physical condition [2,3], good neuromuscular coordination and balance, harmony in movements [4,5] and learning of fighting skills. The development of neuromuscular coordination is a prerequisite for learning, refinement, stabilization and application of Taekwondo skills and the relative execution techniques [6,7]. High-level competitors in this sport are focusing on their coordination improvement, which could increase training effectiveness for their competence [8].

European studies highlighted a decrease in the ability to perform coordinated movements in pre-school and school children [9]. This poor coordination performance might be due to a decrease in quality of physical activity that children actually experience [10].

Taekwondo techniques are effective successions of movements to solve specific problems in sporting situations [11]. The expression of an action between two opponents (e.g., an attack of the one or counterattack of the other) consists of a series of movements appropriate for the particular space, time, distance and opponents’ stance [12,13]. The requirements of Taekwondo techniques include the development of the perfect execution of forms [14], the ability to move in the competition area with speed, agility and balance [15], to execute offensive and defensive techniques with precision, and speed and power and to execute special techniques (breakings, skills) with accuracy and power. Balance plays a crucial role in almost all sports disciplines, inducing specific posture adaptations associated with the right muscle involvement to execute specific movements [16]. The relationship between skill and the development of movement in an environment that is constantly changing and unpredictable requires different adaptations of the technical ability. Physical condition and technique learning are the prerequisites of sports training, while the solution of a problem in a sporting situation depends on both movement automatization and executive cognitive functions development. Cognitive functions allow the athlete to think before acting; these include the inhibitory skill, which controls impulsive or automatic responses; memory, to retain and manipulate information; and cognitive flexibility, which reflects on the possible consequences of specific actions, creating suitable responses [17]. In other words, athletes need an efficient mental engagement by which they recognize stimuli and new information to master new skills, suitable for specific situations [18]. Childhood is an important and sensitive period for this typology of cognitive development [19]. There are many studies on Taekwondo physical and technical–tactical performance both in young and adult athletes [20,21,22], while fewer studied the cognitive and coordinative benefits in performing taekwondo at an early age.

Therefore, this study aims to verify if taekwondo practice enhances visual reaction time, speed and precision in performing technical movements such as kicking and anticipatory skills in agility performance in children, comparing different groups based on age, gender and technical level.

## 2. Materials and Methods

### 2.1. Experimental Approach

To compare the visual reaction time, speed and precision in performing technical movements such as kicking and anticipatory skills in agility performance between groups, all participants underwent three testing sessions in three non-consecutive days, at the same time of the day, in the following order: the ruler drop test [23], the hexagonal test [24] and the target kick test [25]. For these tests, the dominant leg was defined as the leg used in order to manipulate an object or to lead out in movement. Instead, the non-dominant leg was defined as the leg which performs the stabilizing or supporting role [26]. Participants were requested to avoid strenuous activities for at least 48 h before each test. All athletes were familiarized with the testing procedures 1 week prior to the day of the tests.

### 2.2. Participants

One hundred and fifteen children (aged 7.76 ± 1.71 years), 83 males and 32 females, were recruited from several Taekwondo Greek clubs. All participants had regular training practice that included 2 sessions per week, lasting approximately 90 min. The inclusion criteria were no injuries in the last 2 months and no use of drugs that could influence the correct execution of the tests.

The recruited sample was divided in categories based on:Age: under 8 (5–7 years), under 10 (8–9 years) and under 12 (10–11 years).Gender: males and females.Technical level: according to the belt color achieved by the subject, high (1 Dan–5 Kup) and low (6 Kup–9 Kup).

For the comparison between genders on coordination ability, the analysis was performed only on 32 males and 32 females to guarantee the sample homogeneity. The 32 males were chosen following these inclusion criteria: to have the same age and technical level (according to the belt color achieved by the subject) of the 32 females. Sample characteristics are shown in Table 1.

After being informed about the study procedures, risks and aims, the participants’ parents signed the written consent. The study was designed and conducted in accordance with the Declaration of Helsinki and approved by the bioethical local committee of the National and Kapodistrian University of Athens.

### 2.3. The Ruler Drop Test

The RDT aims to evaluate visual reaction time. Materials needed are a one meter ruler, a normal height desk and one assistant. The athlete stands bending over the edge of the desk while the dominant hand forearm is flat on the edge of the desk. The dominant hand was assessed according to Brown et al. [27]. The opposite hand must assist the dominant hand in order to keep the hand stationary; the athlete’s fingers are extended and their thumb is pointing upward. The ruler is held perpendicular to the floor so that the end is even with the top of the index finger and is positioned 4–6 cm from the palm so that when the subject brings the thumb and fingers together, it will be caught. The subject is instructed to watch their hand and the end of the ruler and he/she has to catch the ruler as quickly as possible by moving only their fingers and thumb when they see the ruler move. The assistant informs them that the ruler will be dropped 1–5 s after the start signal. Finally, the assistant drops the ruler and records at the top of the index finger the centimeters. The subject was allowed one practice test and the tester made any corrections needed. The best of three attempts was considered for the analysis. This test has a good test-retest reliability (ICC = 0.744) [28].

### 2.4. The Hexagonal Test

The objective of the HT is to evaluate the athlete’s agility. Materials needed in order to properly conduct this test are a 66 cm sided hexagon marked out on the floor (Figure 1), a stopwatch and one assistant.

The athlete stands in the middle of the hexagon, facing line A throughout the whole procedure of the test. On the command “Go”, the watch is started and the athlete jumps with both feet over line B and back to the middle, then over line C and back to the middle, then line D and so on, until he/she jumps over line A and back to the middle again thus counting as one circuit. After the completion of three circuits, the watch is stopped and the time recorded. After a 30 min rest, the test is repeated and the average of the two recorded times is determined. If someone jumps the wrong line, or lands on a line, then the test has to be restarted. The hexagonal test has a good interclass correlation (ICC) ranged between 0.86 and 0.95 [29].

### 2.5. The Target Kick Test

The objective of the TKT is to evaluate the athlete’s visual reaction time by reacting with a roundhouse kick and his ability to perform maximal number of kicks possible. Materials needed in order to properly conduct this test are a kick target, a counter and two assistants. The athlete stands in the basic fighting stance facing the assistant who holds the target at their waist height. The dominant leg is in the rear and with the command of the counter the subject starts kicking with his rear leg for 10 s landing every time the leg to the rear. The dominant leg was assessed according to Schneiders et al. [30]. The assistant holding the target counts only the successful kicks. The target kick test has a very good test-retest reliability (ICC = 0.85) [25].

### 2.6. Statistical Analyses

Data analysis was performed using SPSS Statistics 23 (IBM) software. The Kolmogorov–Smirnov test was applied, before analysis, to test the normal distribution of data. A descriptive analysis of the sample was carried out. Continuous values are expressed as mean and standard deviation (SD).

Due to the normal distribution of data, a multivariate analysis of variance (MANOVA) test was performed to assess the differences amongst the performer test results as dependent variables. Gender, age and technical level were considered as independent variables. Post-hoc comparisons were performed using the Bonferroni test. Furthermore, 95% of confidence intervals (CIs) for the differences were reported. The alpha test level for statistical significance for all variables was set at 0.05. The Cohen’s d effect size was calculated for statistically significant differences: values below 0.2 and 0.49 were considered the small effect; 0.5 and 0.79 the medium effect; ≥0.8 the large effect [31].

## 3. Results

### 3.1. Age Differences

The MANOVA test showed significant differences for age in the hexagonal test (F = 15.644; df = 2; *p* < 0.001), where the under 12 group showed a lower time of execution than both the under 8 (*p* < 0.001) and under 10 groups (*p* = 0.038) and the under 10 group showed a lower time of execution than the under 8 one (*p* = 0.006). Significant differences were found in the target kick test (F = 10.678; df = 2; *p* < 0.001), where the under 12 group showed a higher score than both the under 8 (*p* < 0.001) and under 10 groups (*p* = 0.034). No significant differences were found for the ruler drop test. The results are shown in Table 2.

### 3.2. Gender Differences

Significant difference was found regarding gender in the target kick test, where females showed a higher score than males (*p* = 0.033). No significant differences between females and males were found in the ruler drop and hexagonal tests. The results are shown in Table 3.

### 3.3. Technical Level Differences

Significant differences were found for technical level, where the high-level group showed better scores in all the tests than the low-level one (ruler drop test, *p* = 0.047; hexagonal test, *p* = 0.022; target kick test, *p* = 0.035). The results are shown in Table 4.

### 3.4. Comparison between Technical Level, Gender and Age

Significant difference was found regarding the comparison between technical level and gender, where low-level females showed better scores than low-level males in the target kick test (*p* = 0.013). No significant differences were found regarding gender in the high-level group. The comparison between technical level and age showed no significant difference between the under 12 low-level group and under 8 high-level one in the target kick test (*p* = 0.058). No significant differences between technical level and gender and between technical level and age were found in the ruler drop test. The results are shown in Table 5.

## 4. Discussion

The study aimed to evaluate age, gender and technical level influence on the several coordination skill developments of young athletes engaged in Taekwondo. Regarding age, the main finding of the study was that no significant differences amongst age groups were found in the ruler drop test assessing the simple reaction time, while agility and precision, assessed by the hexagonal test and the target kick test, respectively, improved with age. It is well known that coordinative abilities improve with maturation of motor skills and nervous system [32] and that coordination is positively influenced by experience in extended sports activity engagement. Therefore, it was expected that older athletes achieved better results [33]. Several studies have shown that motor proficiency develops with age [34,35]. In particular, agility, assessed by the hexagonal test, represents a complex psychomotor skill, involving not only strength and speed, but also balance, coordination and the ability to react to a change of the environment [36]. Several studies highlighted the relation between age and agility, coordination and strength [37,38]. One more study [39] showed that agility development requires a high degree of neuro-muscular specificity, typical of any sport, therefore long-term training may provide this competence. Speed and precision of the kick depend on different parameters, such as foot take-off velocity, acceleration and joint angle control, as well as muscle strength and balance, that influence the kicking techniques [40] and that develop with age and experience. Consequently, motor competence has to be learned, practiced and developed [41] and, in particular, well-structured and planned extracurricular activities have a positive impact on motor development [42].

The reaction time represents a reliable measure to assess the capacity of the cognitive system to process information [43]. In this study, the ruler drop test assessed only the simple reaction time, defined as the time interval between when a stimulus appeared and the given response [44]. No differences amongst different age groups were found in this test. Similar results [45] were found in children between 6–8 and 10–12 years old, but significant differences in the 8–10 years group. Taekwondo involves both simple and complex reaction time. Complex reaction time requires the identification and selection of a response to various stimuli and is substantially influenced by sports experience [46]. Nevertheless, while complex reaction time can be effectively trained [47,48], allowing the development of a wide variety of sports actions during maturation and sports career progression [49,50], simple reaction time is genetically-based and less trainable but strongly related to selective attention and concentration. Consequently, simple reaction time, assessed at an early stage, could be considered a good predictor of Taekwondo performance.

Regarding gender, no significant differences were found in the administered tests, except for the target kick test, in which females achieved better results than males. Gender was not a predictor of skill proficiency in childhood and differences between genders showed mixed results [51]. Previous studies concerning gender differences in youth sport found that both boys’ and girls’ performance was quite similar [52,53]. Prior to puberty, physical characteristics of boys and girls are similar except for some small differences in body composition, length of upper limbs, strength and range of motion. Prolonged and delayed pubertal development might be due to sport training intensity [54]. Throughout the preschool age, females provide better performances in tasks involving coordinative abilities, such as bilateral balance coordination [34], standing on preferred leg on floor and visual-motor control [55] and develop better mental alertness than males [56]. These good prerequisites might explain better performances in the target kick test. This result showed the need to dedicate more time taking into consideration the different technical levels, according to the belt color achieved by the participants; the expected results were confirmed: high-level athletes reached significantly better results than low-level ones. Matching the technical level with the other sample characteristics, interesting results were found in the target kick test, the under 8 of higher technical level reached similar scores to low levels under 12. This finding highlighted that talent and predisposition to a particular sports activity linked to a well-planned training may lead to a motor proficiency comparable to that reached by older athletes and better than same-age athletes. Considering that talent in sport could be defined as an athlete who carries out greater performance compared to same-age athletes, due to his/her exceptional natural ability, these tests would be effective enough to help coaches make appropriate choices in selecting athletes in early stages [57]. No significant results for different technical level between under 12 groups were found. It could be explained considering that high level in Taekwondo, recognized by Kup degrees, is characterized by several and specific technical–tactical aspects and fighting situations under high unpredictability, that are not detectable using these tests. It may be concluded that these tests are not suitable and selective for this age range. Another topic which emerged from these data was that the significant differences, detected between low-level males and females in the target kick test, were lost when considering the athletes’ high technical level. Taking into consideration that identification and selection of elite athletes in Taekwondo are mainly based on kicking ability and dynamic footwork, high-level male athletes may improve this skill over the years similarly to females [58,59].

### Limitations

The findings of this study should be viewed in the context of the following limitations: the sample size of female athletes compared to males; only three tests were used to evaluate the coordinative abilities, giving the possibility to test only few abilities referred to Taekwondo; finally, the coordinative skills level was assessed by field tests.

## 5. Conclusions

While the conditional profile of Taekwondo athletes is widely documented, little or nothing was published in sports studies on motor coordination, especially related to young athletes. Taekwondo puts young athletes in the process of developing complex cognitive skills, such as thinking, identifying cause-and-effect relationships and developing creative thinking and planning skills [59], offering training chances in the form of motor multifaceted experiences. Additionally, it provides a variety and quantity of movements and demands in regard to coordination capabilities, using specific training methods in order to maximize performance. Children who master skilled movements might feel more competent in moving, perceiving more enjoyment in moving. Moreover, children with better coordination abilities are those having higher chances of further progress in this sport. Especially during the developmental ages, coordination and motor proficiency must be monitored to effectively plan training programs for young Taekwondo athletes. In this study, the efficacy of three tests, aiming to assess the athletic progress based on age, gender and technical level, was analyzed. The ruler drop test is a good tool to select reaction time ability, independently of age, gender and technical level. Considering that the reaction time allows to reduce the decision-making time in complex motor tasks, it could be considered a predictor of talent in Taekwondo. The hexagonal test, assessing agility, which involves strength, speed, balance and coordination, could be a good test, feasible for different ages and technical levels. The target kick test monitors the most peculiar skill in Taekwondo, which develops with age and is related to technical level.

Considering the results of this study, the assessment of technical–tactical capabilities of young Taekwondo athletes could help to plan and optimize effective youth programs and to find strategies to promote the potential of children. A well-managed sports activity means for children an improvement of coordinative abilities, especially needed in this sensitive age. In particular, in this period in which there is a trend of general decrease of the ability to perform coordinative movements in younger generations, the role of sport becomes fundamental. Moreover, the added value of sports activity to increase self-efficacy and enjoyment in children has to be highlighted, considering that these feelings are strong predictors of adolescent and adult engagement in active lifestyles. These improved self-efficacy and enjoyment are strong predictors of adolescent and adult engagement in active lifestyles.

## Figures and Tables

**Figure 1 ijerph-18-05998-f001:**
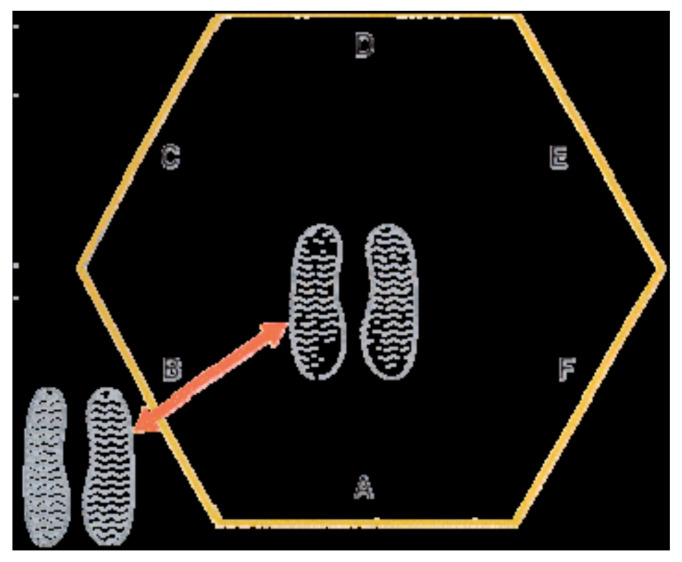
Hexagonal test obstacle.

**Table 1 ijerph-18-05998-t001:** Sample characteristics.

Variable	n
Total	115
Gender	
Male	83
Female	32
Age	
Under 8 (5–7)	55
Under 10 (8–9)	37
Under 12 (10–11)	23
Technical level	
High (1 Dan–5 Kup)	52
Low (6 Kup–9 Kup)	63

**Table 2 ijerph-18-05998-t002:** Age differences in the motor coordination tests (n = 115).

Variable	Groups		Mean ± SD		Standard Error	95% CI		*p*-Value	Effect Size
						Low	High		
Ruler drop test	Under 8	Under 10	31.80 ± 11.36	30.48 ± 7.09	2.071	−3.71	6.36	1.000	
	Under 8	Under 12	31.80 ± 11.36	30.29 ± 9.18	2.587	−6.10	6.48	1.000	
	Under 12	Under 10	30.29 ± 9.18	30.48 ± 7.09	2.587	−6.48	6.10	1.000	
Hexagonal test	Under 8	Under 10 *	25.63 ± 8.86	20.39 ± 7.82	1.656	1.21	9.27	0.006 *	0.627
	Under 8	Under 12 *	25.63 ± 8.86	15.15 ± 4.02	1.934	5.78	15.19	<0.001 *	1.523
	Under 12 *	Under 10	15.15 ± 4.02	20.39 ± 7.82	2.068	−10.27	−0.22	0.038 *	0.843
Target kick test	Under 8	Under 10	4.85 ± 2.07	5.98 ± 2.44	0.522	−2.39	0.15	0.104	
	Under 8	Under 12 *	4.85 ± 2.07	7.65 ± 3.24	0.610	−4.28	−1.31	<0.001 *	1.030
	Under 12 *	Under 10	7.65 ± 3.24	5.98 ± 2.44	0.652	0.09	3.27	0.034 *	0.582

Under 8: 5–7 years; Under 10: 8–9 years; Under 12: 10–11 years. * Significant differences.

**Table 3 ijerph-18-05998-t003:** Gender differences in the motor coordination tests (n = 64).

Variable	Gender	n	Mean ± SD	*p*-Value	Effect Size
Ruler drop test	Male	32	32.59 ± 10.41	0.689	-
	Female	32	31.61 ± 9.05		
Hexagonal test	Male	32	21.38 ± 8.38	0.954	-
	Female	32	21.26 ± 7.91		
Target kick test	Male	32	5.03 ± 3.04	0.033 *	0.547
	Female *	32	6.53 ± 2.41		

* Significant differences.

**Table 4 ijerph-18-05998-t004:** Technical level differences in the motor coordination abilities (n = 115).

Variable	Technical Level	n	Mean ± SD	*p*-Value	Effect Size
Ruler drop test	High *	52	29.10 ± 9.06	0.047 *	0.378
	Low	63	32.70 ± 9.95		
Hexagonal test	High *	52	19.81 ± 8.63	0.022 *	0.434
	Low	63	23.53 ± 8.52		
Target kick test	High *	52	6.35 ± 3.05	0.035 *	0.395
	Low	63	5.30 ± 2.20		

* Significant differences.

**Table 5 ijerph-18-05998-t005:** Comparison between technical level, gender and age in the hexagonal test and target kick test (n = 115).

Variable	Groups	Mean ± SD	*p*-Value	Effect Size
Hexagonal test	Female HL *–Female LL	17.19 ± 7.21–23.43 ± 7.52	0.030 *	0.847
	Under 8 HL–Under 10 HL *	24.81 ± 9.28–16.19 ± 5.49	0.002 *	1.131
	Under 8 HL–Under 12 HL *	24.81 ± 9.28–14.51 ± 4.51	0.001 *	1.412
	Under 8 LL–Under 12 LL *	26.27 ± 8.63–15.73 ± 3.61	<0.001 *	1.593
	Under 12 LL *–Under 10 LL	15.73 ± 3.61– 23.97 ± 7.83	0.014 *	1.352
	Under 8 HL–Under 12 LL *	24.81 ± 9.28–15.73 ± 3.61	0.003 *	1.290
	Under 10 HL *–Under 8 LL	16.19 ± 5.49–26.27 ± 8.63	<0.001 *	1.394
	Under 10 HL *–Under 10 LL	16.19 ± 5.49–23.97 ± 7.83	0.002 *	1.151
	Under 12 HL *–Under 8 LL	14.51 ± 4.51–26.27 ± 8.63	<0.001 *	1.708
	Under 12 HL *–Under 10 LL	14.51 ± 4.51–23.97 ± 7.83	0.001 *	1.481
Target kick test	Male LL–Female LL *	4.38 ± 2.13–6.14 ± 2.26	0.013 *	0.801
	Under 8 HL–Under 12 HL *	5.29 ± 2.37–8.45 ± 4.01	0.011 *	0.959
	Under 8 LL–Under 12 LL *	4.52 ± 1.77–6.92 ± 2.27	0.003 *	1.179
	Under 10 HL *–Under 8 LL	6.47 ± 2.62–4.52 ± 1.77	0.004 *	0.872
	Under 12 HL *–Under 8 LL	8.45 ± 4.01–4.52 ± 1.77	<0.001 *	1.268
	Under 12 HL *–Under 10 LL	8.45 ± 4.01–5.55 ± 2.26	0.015 *	0.891

HL: high level; LL: low level; Under 8: 5–7 years; Under 10: 8–9 years; Under 12: 10–11 years old. * Significant differences.

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
