# Peer review of "The Impact of Age, Gender and Technical Experience on Three Motor Coordination Skills in Children Practicing Taekwondo"

_ijerph, 2021, doi:10.3390/ijerph18115998_

Round 1

Reviewer 1 Report

The aim of this study was to investigate the influence of age, gender and technical level in motor coordination abilities of children engaged in Taekwondo.

In general terms, the objective and procedure of the work is perfectly understood, although there are some aspects that need to be clarified.

In the first place, the authors have passed 3 tests to a sample of athletes who practice Taekwondo, but the title refers to "the Motor Coordination Abilities". What is really done in this study is to determine the incidence / relationship of sex, age and technical level of athletes who practice taekwondo on (three) specific motor control tests, so it is suggested to change and "adjust" the title of the work .

1. In the title and in the abstract, reference is made to the analysis by sex and age, but then we see that the technical level is also incorporated, why is it excluded in the previous sections?

2. "For the comparison between genders, on coordination ability, the analysis was performed only on males and 32 females to guarantee the sample homogeneity (age and technical level). I do not understand why this decision was made and I do not understand the justification in statistical terms.

3. In test 1 and in test 3 reference is made to the dominant hand and the dominant leg. How has the concept of dominant hand and leg been established? How was hand and leg dominance determined?

4. The presentation of the results is quite confusing, so a clearer, more orderly and complete presentation is recommended.

5. In the conclusions, it seems that the authors focus more on justifying the usefulness of the tests used than on concluding aspects of their research, so I recommend a reflection on this section.

6. The authors should include a section on limitations since, in addition to having many, none have been exposed.

7. Language review is recommended.

8. General revision of punctuation marks in the text (for example line 60) and end of some lines (for example line 57) is recommended.

Reviewer 2 Report

Thank you for the opportunity to review this manuscript, which focuses on the important health issue of fostering children’s athletic ability. Opportunities for improvement are suggested below, which mostly focus on clarifying the need for and implications of the study.

Introduction

*In general, the introduction could be improved by making the case for why the study is needed. What knowledge gaps is the study addressing? Why is knowing the coordination level of children involved in an activity (in this case TKD) important? In later parts of the manuscript, the authors discuss implications of their study for selecting elite TKD athletes during childhood, but many readers (such as myself) may want to know about broader child health and public health implications.

*Line 40 is a one sentence paragraph. The authors may want to incorporate into preceding paragraph.

*The paragraph in lines 47-73 is very long, and reads morso as an advertisement for TKD rather than content that puts the study into context.

Methods

*What is the rationale for the age group divisions that were chosen? Is there a developmental rationale?

Discussion

*I would like to read more about how the study relates to issues of interest to IJERPH’s readership (environment and public health). Currently, the focus of the discussion is narrowly focused on athletic performance in one sport (TKD). What are the broader implications for public health and child health promotion? What do study results mean for parents/caregivers, teachers, coaches, or individuals who make policies related to youth programming? What future research is needed? The last sentence of the discussion briefly alludes to this when stating “Considering the results of this study, the assessment of technical-tactical capabilities of young Taekwondo athletes could help to plan and optimize effective youth programs and to find strategies to promote the potential of children”, but it is not addressed elsewhere and I would like to read more.

*The paper is missing a limitations section (e.g., small convenience sample).

Round 2

Reviewer 1 Report

In my opinion, most of the considerations made have been addressed.
However, I would like two issues to be clarified:

a) In point 2 of my report the authors respond that "Due to the different number of male and female participants (83 males and 32 females), we selected 32 males with the same age and technical level of females, in order to exclusively focus the analysis on the gender differences, perform a more correct statistical analysis. Comparing two sample different for number of subjects the standard error become to major and consequently the t value -all other things being equal - become less significant ".
The authors' decision is perfectly understandable but they should explain how the selection of the 32 men from an initial sample of 83 was carried out (inclusion criteria, etc.).

b) In point 3 of my report the authors respond "Children involved in this study practice Taekwondo by several years; in addition, in this developmental stage they had already structured a dominant hand and leg".
It is not my intention to go into discussing this argument, but I do not agree, since one thing to have acquired sports lateralization and another is to define what the authors understand by dominant hand or leg. For example, in a round kick which is the dominant leg? Support leg or free leg? For example, the athletes analyzed only use one hand to hit? For example, athletes always use the same hand to perform different strokes? That is, if the authors are going to use the dominant hand or dominant leg variable for analysis, they must define and evaluate it with specific criteria.

Reviewer 2 Report

Thank you for the opportunity to review this manuscript. The reviewers have addressed prior concerns. Two minor concerns remain:

*The authors have added two one-sentence paragraphs (line 226, line 314). Can the expand, or incorporate into prior paragraphs?

*The limitations section the authors have added is in bullet/list format. Can the the authors write in narrative format? (line 277)
